# Focality-Oriented Selection of Current Dose for Transcranial Direct Current Stimulation

**DOI:** 10.3390/jpm11090940

**Published:** 2021-09-21

**Authors:** Rajan Kashyap, Sagarika Bhattacharjee, Ramaswamy Arumugam, Rose Dawn Bharath, Kaviraja Udupa, Kenichi Oishi, John E. Desmond, S. H. Annabel Chen, Cuntai Guan

**Affiliations:** 1School of Computer Science and Engineering, Nanyang Technological University, 50 Nanyang Avenue, Singapore 639798, Singapore; ARUMUGAM004@e.ntu.edu.sg; 2School of Social Sciences (SSS), Nanyang Technological University, Singapore 639818, Singapore; bhattacharya.sagarika7@gmail.com (S.B.); annabelchen@ntu.edu.sg (S.H.A.C.); 3Centre for Research and Development in Learning (CRADLE), Nanyang Technological University, Singapore 637460, Singapore; 4Department of Neuroimaging and Interventional Radiology, National Institute of Mental Health and Neurosciences, Hosur Road, Bangalore 560029, India; drrosedawn@nimhans.kar.nic.in; 5Department of Neurophysiology, National Institute of Mental Health and Neurosciences, Hosur Road, Bangalore 560029, India; kaviudupa.nimhans@nic.in; 6The Johns Hopkins University School of Medicine, Baltimore, MD 21205, USA; koishi@mri.jhu.edu (K.O.); jdesmon2@jhmi.edu (J.E.D.); 7Lee Kong Chian School of Medicine (LKC Medicine), Nanyang Technological University, Singapore 637553, Singapore; 8National Institute of Education, Nanyang Technological University, Singapore 637553, Singapore

**Keywords:** transcranial direct current stimulation (tDCS), realistic volumetric approach-based simulator for transcranial electric stimulation (ROAST), systematic approach for tDCS analysis (SATA), current dose, individualized tDCS, age and sex difference

## Abstract

**Background:** In transcranial direct current stimulation (tDCS), the injected current becomes distributed across the brain areas. The objective is to stimulate the target region of interest (ROI) while minimizing the current in non-target ROIs (the ‘focality’ of tDCS). For this purpose, determining the appropriate current dose for an individual is difficult. **Aim:** To introduce a dose–target determination index (DTDI) to quantify the focality of tDCS and examine the dose–focality relationship in three different populations. **Method:** Here, we extended our previous toolbox i-SATA to the MNI reference space. After a tDCS montage is simulated for a current dose, the i-SATA(MNI) computes the average (over voxels) current density for every region in the brain. DTDI is the ratio of the average current density at the target ROI to the ROI with a maximum value (the peak region). Ideally, target ROI should be the peak region, so DTDI shall range from 0 to 1. The higher the value, the better the dose. We estimated the variation of DTDI within and across individuals using T1-weighted brain images of 45 males and females distributed equally across three age groups: (a) young adults (20 ≤ x ˂ 40 years), (b) mid adults (40 ≤ x ˂ 60 years), and (c) older adults (60 ≤ x ˂ 80 years). DTDI’s were evaluated for the frontal montage with electrodes at F3 and the right supraorbital for three current doses of 1 mA, 2 mA, and 3 mA, with the target ROI at the left middle frontal gyrus. **Result:** As the dose is incremented, DTDI may show (a) increase, (b) decrease, and (c) no change across the individuals depending on the relationship (nonlinear or linear) between the injected tDCS current and the distribution of current density in the target ROI. The nonlinearity is predominant in older adults with a decrease in focality. The decline is stronger in males. Higher current dose at older age can enhance the focality of stimulation. **Conclusion:** DTDI provides information on which tDCS current dose will optimize the focality of stimulation. The recommended DTDI dose should be prioritized based on the age (>40 years) and sex (especially for males) of an individual. The toolbox i-SATA(MNI) is freely available.

## 1. Introduction

Transcranial direct current stimulation (tDCS) is a noninvasive brain stimulation technique that could alleviate symptoms of several neurological and psychiatric brain disorders [1,2,3]. A conventional tDCS setup consists of an anode and cathode placed over the scalp (referred to as a ‘montage’) with a low intensity of current (~1–3 mA) being injected to stimulate the target region of interest (ROI) [4,5]. However, the injected current becomes diffused in the intermediary regions of the brain and might not effectively stimulate the target ROI with the desired intensity [6,7]. Computational models that predict the pattern of current flow across the brain of an individual are used to optimize the tDCS stimulation parameters [8,9,10,11,12,13,14]. The amount of injected current (referred as the ‘current dose’) plays an important role in the dispersal of the stimulation’s intensity across the brain regions [15,16]. The distribution may vary from person to person and within a person based on the quantity of the dose [17,18,19]. Therefore, the selection of the optimal current dose for an individual’s brain that could sufficiently stimulate the target ROI while minimizing the current in non-target ROIs is important.

In recent years, there has been a growing interest in the individualization of the current dose [15,16,20]. It has been reported that varying the current intensity on the scalp for each individual can reduce the interindividual variability in the electric field intensity (or current density) at the target ROI [20]. The current dose calculated through inverse modelling of the tDCS-induced electric field at the target ROI correlates with the motor thresholds generated by transcranial magnetic stimulation [15]. In a recent tDCS experiment using a frontal montage and a 2 mA (fixed) current dose, individuals with a high current density at the target ROI (left dorsolateral prefrontal cortex) were found to have stronger improvements in working memory compared to those with a low current density [21]. They also showed that individualizing the current dose by fixing the desired current density at the target region can maximize the benefits of tDCS [21]. Though the models are a step towards individualizing the current dose, they do not consider the spread of the field to intermediary (non-target) regions. The current flow in the intermediary regions have a vital role to play in determining the outcome of tDCS [6,12,22,23,24,25]. It has been found that some brain regions may act as conduits, clustering most of the current to a specific location that can deter the intensity of the stimulation expected at the target ROI [6,26]. At this point, it is important to mention that other stimulation techniques (like peripheral nerve stimulation) are also intended to increase the stimulation intensity at target region while minimizing the stimulation received at non-target regions [27,28,29,30]. With tDCS, poor focality in stimulating the target region has constrained its efficacy. Therefore, the approaches to individualize the current dose [31,32,33,34] should consider the focality of stimulation in order to recommend the optimal intensity of input current.

In our previous work, we developed an individual-Systematic-Approach-for-tDCS-Analysis (i-SATA) toolbox [35] that estimates the average current density received by target ROIs and intermediary regions of an individual’s brain after a montage has been simulated in a realistic volumetric approach-based simulator for transcranial electric stimulation (ROAST) toolbox [10]. The ROAST-simulated current density in the ROIs has been found to be strongly correlated with electrophysiological measurements performed in vivo [9]. Integrated with ROAST, the i-SATA toolbox can be applied on an individual brain to reverse-calculate the current dose that can be used to stimulate the target ROI with the desired intensity [35]. This was performed based on the assumption that electric field intensity at the target ROI increases linearly with increase in the current dose by following the procedure laid down by Evans et al. [20]. Since we will be using it throughout the study, it will be helpful to familiarize our readers with an example. Suppose the calculated current density at the target ROI is 0.25 mA/m^2^ when 1 mA of current is applied on the scalp. To achieve a desired density of 0.5 mA/m^2^ at the target ROI, the required dosage (individualized) can be reverse-calculated as $Individualised\;dose=\left(\frac{Desired\;Current\;Density}{Actual\;Current\;Density}\right)\times Fixed\;dose\;[i.e.,\left(\frac{0.5}{0.25}\right)\times 1$ = 2 mA].

In i-SATA, we used the Talairach client toolbox [36] to map an individual brain to the Talairach atlas space [37]. Another widely used brain template that provides detailed stereotaxic information on the location and variability of cortical areas is provided by the Montreal Neurological Institute (MNI) reference space [38,39,40,41,42]. Simon et al. [43,44,45] had developed the SPM anatomy toolbox that integrates the cytoarchitectonic maps in the MNI space. Here, we leveraged the potential of the SPM anatomy toolbox to extend i-SATA to the MNI space. The extended i-SATA(MNI) toolbox, which integrates the SPM anatomy toolbox with i-SATA, will enable researchers to visualize the comprehensive overview of the current density distribution across the cortex (target and intermediary regions) in the MNI space.

With i-SATA(MNI), we introduce the *dose–target determination index* (DTDI), a simple estimate that will quantify the focality of stimulation and facilitate the selection of optimal current dose required to stimulate the target ROI in an individual’s brain. A similar metric defined as the ‘selectivity index’ that measures the recruitment of the targeted region compared to other non-targeted regions is used to quantify the effectiveness of peripheral nerve stimulation [27,28,29,30]. For tDCS, the DTDI will aim to provide a comprehensive overview of the intensity of the stimulation received by the target ROI and intermediary regions after a montage has been postprocessed in i-SATA(MNI). To explain DTDI, we will use the montage with an anode positioned at F3 and a cathode at the right supraorbital (RSO) (referred to as F3-RSO). The montage has been shown to stimulate the left middle frontal gyrus [22,25] and is effective for depression [3,22,46] and working memory [47]. To make it easy for our readers to interpret how DTDI facilitates selection of the current dose, we will show the interindividual as well as the intraindividual variation in the index by uniformly increasing the current dose. Finally, we will evaluate the variation in DTDI by the age and sex of individuals. The purpose will be to explore if dose selection should be prioritised for any category (age and sex) of individuals.

## 2. Methods

### 2.1. Data

We obtained the T1-weighted (T1WI) magnetic resonance image (MRI) of the brain of 90 age–sex matched healthy individuals (45 male) from Cambridge Centre for Ageing and Neuroscience (Cam-CAN) study (available at http://www.mrc-cbu.cam.ac.uk/datasets/camcan/, accessed on 21 October 2020 [48,49]). This study was approved by the local ethics committee, Cambridgeshire 2, Research Ethics Committee (reference: 10/H0308/50). In this study, the Cam-CAN team recruited adult participants (aged 18–87 years old) in three stages that comprised of a home-based interview (stage one), followed by an evaluation of their health status (stage two). Subjects that were cognitively healthy (determined by a mini-mental state exam (MMSE) score ≥ 27), who met hearing, vision, and English language ability criteria, and who were free of MRI contraindications and neurologic or psychiatric conditions were recruited for stage three. In stage three, multimodal data (functional and structural MRI, magnetoencephalography, and behavioural) were collected from each participant. The T1WIs were collected from a 3T Siemens TIM Trio scanner with a 32-channel head coil using an MPRAGE sequence, TR = 2250 milliseconds (ms), TE = 2.99 ms, flip angle = 9°, Voxel size = 1 × 1 × 1 mm^3^, FOV = 256 × 240 × 192 mm^3^, GRAPPA: 2; TI: 900 ms. We selected 90 T1WIs from the following three age groups with 30 individuals (15 right-handed males and females) in each group:(a) young adults (20 ≤ x ˂ 40 years), (b) mid adults (40 ≤ x ˂ 60 years), and (c) older adults (60 ≤ x ˂ 80 years) were selected. The equal grouping across the three groups would allow an evaluation of the relationship of tDCS current dosage with sex and age.

### 2.2. Preprocessing with ROAST

We simulated the montage F3-RSO with the electrode size 5 × 5 cm^2^ (Figure 1A). For each individual MRI, the montages were simulated for three current doses of 1 mA, 2 mA, and 3 mA. In total, 270 simulations were performed in ROAST (Total = 90 MRI × 3 current doses = 270) [10]. Default conductivity values of the tissues (white matter (default 0.126 S/m); grey matter (default 0.276 S/m); cerebrospinal fluid (default 1.65 S/m); bone (default 0.01 S/m); skin (default 0.465 S/m); air (default 2.5 × 10^−14^ S/m); gel (default 0.3 S/m); electrode (default 5.9 × 10^7^ S/m)) were used for each MRI that we simulated in ROAST. The ROAST simulation outputs the locations (x, y, and z coordinates) of the brain regions and the current density (mA/m^2^) value at each location in the native space.

### 2.3. i-SATA(MNI)

The i-SATA(MNI) is similar to i-SATA (available for download at https://doi.org/10.21979/N9/5W3RIM, [35]) except for the atlas space. In short, for each montage simulated in ROAST, i-SATA extracts the location (x, y, and z coordinates) of all the points in the cortex to detect the location of three anatomical landmarks (anterior commissure, posterior commissure, and mid-sagittal) using the acpcdetect toolbox [35,50]. With these landmarks, the individual’s native space was mapped to the reference space (i.e., the Talairach atlas space) using the fieldtrip toolbox [51] followed by the Talairach client toolbox [36]. Details on the methodology and application can be obtained from previous works [11,35,52]. For i-SATA(MNI), instead of the Talairach atlas space, we mapped the outputs (x, y, and z coordinates) to the MNI reference space using the SPM anatomy toolbox [43,44,45]. The SPM anatomy toolbox has an option for using the gyri/sulci-based labelling system wherein the automated anatomical labeling atlas with 116 regions outlined on the Colin27 brain template is implemented (for details, see [53]). The i-SATA(MNI) extracts and uses the labels provided by this atlas for assigning the cortical and subcortical region corresponding to each location. A detailed explanation on the nomenclature of the delineated regions can be found at [53]. We developed i-SATA(MNI) using SPM12 (revision 6470, available at https://www.fil.ion.ucl.ac.uk/spm/software/spm12/) that has the SPM Anatomy toolbox (version 2.2b) inbuilt into the framework. The magnitude of current density corresponding to each location (voxel) was then used to obtain the average magnitude of the current density received by each cortical region of the brain. This will provide an estimate of the current density induced in the target and intermediary region due to tDCS. As an example, we will postprocess the standard MNI 152 averaged head in i-SATA(MNI) for the three current doses (1 mA, 2 mA, and 3 mA) using the F3-RSO montage to show the distribution of the average current density across the cortical regions (Figure 1B–D).

### 2.4. Dose–Target Determination Index (DTDI)

The output of i-SATA(MNI) (i.e., the average current density in the target ROI and the non-target regions) is used to calculate the DTDI for a montage simulated at a current dose. For this, we will find the ROI that has the maximum value of average current density (peak region) amongst all the ROIs. DTDI is then calculated as
$$DTDI=\frac{Average\;Current\;density\;at\;the\;Target\;ROI}{Maximum\;value\;of\;average\;current\;density\;formed\;at\;any\;ROI}$$

DTDI will lie within the range of 0 to 1. An ideal tDCS setup will deliver the maximum intensity of stimulation (average current density) to the target ROI, thereby generating a DTDI value equal to 1. However, the peak intensity may be received at non-targeted ROI. A DTDI value = 0 will indicate no stimulation of the target ROI. For an individual, the current dose for which DTDI is higher should be preferred over other doses. To make this clear, we will estimate the DTDI of three individuals across three current doses. Hypothetically, the value of the DTDI should remain constant across doses, since it is assumed that the current flow in the brain increases linearly with an increase in current intensity [15,16,20,23,54].

### 2.5. Statistical Analysis of Variation in DTDI

All individual MRIs were post-processed in i-SATA(MNI) for the three current doses using the F3-RSO montage to estimate the DTDI’s (Total = 90 MRI × 3 current doses = 270). We show the interindividual and intraindividual variation in the DTDI for both sexes across the three age groups. We performed a three-way mixed ANOVA with age and sex as between subject and dose as within subject factor. Post hoc analyses were performed to further characterize the nature of the main effects and interactions.

### 2.6. Code Availability

The i-SATA(MNI) is a Linux-based MATLAB toolbox integrating acpcdetect v2.0, fieldtrip, and SPM12 (version 6470) with the integrated SPM Anatomy toolbox (version 2.2b). The package can be downloaded at (https://doi.org/10.21979/N9/KWTCWK). A reference manual is also provided to help users run each step with ease.

## 3. Results

### 3.1. Output of i-SATA(MNI) on the Standard Head Model

The montage F3-RSO that we applied to the MNI 152 averaged head model and simulated in ROAST is shown in Figure 1A. The output of i-SATA(MNI), i.e., the distribution of the average current density across the cortical regions, are shown in Figure 1B–D for the three current doses (1 mA, 2 mA, and 3 mA). The average current density in the target ROI (the left middle frontal gyrus) varies linearly with the current dose. Therefore, the DTDI remains constant (at approximately ~ 0.85) across the doses. Of note, similarly to i-SATA [35] and SATA (the standard head model with a graphical user interface is available for download at https://doi.org/10.21979/N9/DMWPZK [11,52,55]), users can visualize the i-SATA(MNI) outputs on the brain surface as well (Figure not shown).

### 3.2. Interpretation of DTDI for Appropriate Selection of Current Dose

For any individual, DTDI can guide the selection of the appropriate current dose that will sufficiently stimulate the target ROI with minimal spread of current to other regions. For interpretation, we have shown the variation of DTDI for three individuals across the three current doses (Figure 2). For the first individual, the current intensity at the target region increases with the increase in dose, and the DTDI remains fairly constant (Figure 2A). This implies that the target ROI will be sufficiently stimulated by any current dose, and that the user can tune it according to the extent of stimulation desired. For the second individual, a low DTDI (0.43) is seen at lower dose (1 mA) suggesting that target ROI is receiving minimal current and non-target regions are receiving most of the current. With the increase in dose, it can be seen that the current intensity at the target ROI is increasing, but fewer regions are receiving a current higher than the target ROI. As a result, the DTDI is increasing with increase in the dose, suggesting that higher current dose should be beneficial (Figure 2B). Finally for the third individual, a decrease in DTDI is seen with the increase in dose (Figure 2C). The drop in DTDI from 1 mA to 2 and 3 mA seems to be due to an increase in current in the right superior parietal lobule at 2 mA and 3 mA only. Although the current intensity at the target ROI increases with the increases in the dose, the maximal amount of current also becomes dissipated to other brain regions. Thus, the conventional way of increasing the current dose to attain the desired stimulation intensity at the target ROI might result in the stimulation of unwanted brain regions (as seen for the superior parietal lobule). For this individual, a lower dose showing higher DTDI can maximize the advantages of stimulation.

### 3.3. Statistical Analysis of Variance in DTDI

The change in DTDI as a function of the dose for males and females across three age groups is shown in Figure 3. The mixed ANOVA revealed a significant main effect of age (F(2, 84) = 43.98, *p =* 8.51 × 10^−14^, effect size = 0.405 (generalised eta squared)), with DTDI significantly decreasing in older adults compared to young adults (*p =* 0.0008). The main effect of sex (F(1, 84) = 12.14, *p <* 7.85 × 10^−4^, effect size = 0.086) and its interaction with age (F(2, 84) = 3.78, *p*= 2.70 × 10^−2^, effect size = 0.05) was also found to be significant. The post hoc analysis shows that females had higher DTDI values than males for both mid (*p* = 2.89 × 10^−6^) and older adults (*p* = 3.18 × 10^−3^). The interaction effect of age and dose was also found to be significant (F(3.34, 140.48) = 7.269, *p* = 7.65 × 10^−5^, effect size = 0.05). In older adults only, the post hoc analysis revealed that there is a significant increase in the DTDI values at 3 mA compared to 1 mA for both males (*p* = 0.01) and females (*p* = 0.02). All the post hoc pairwise comparisons were Bonferroni-corrected. This shows that the focality of stimulation could be enhanced in older adults by increasing the dose.

## 4. Discussion

In this paper, we extended the toolbox i-SATA [35] to the MNI reference space to enable users to obtain the average current density induced at each cortical region of an individual’s brain during tDCS. We then used these values to estimate the DTDI as an objective measure to quantify the focality of stimulation and aid the selection of appropriate current dose for an individual. We demonstrated the utility of DTDI across three subjects and doses (Figure 2). The optimal stimulation of the target ROI was experienced differently across the three subjects: (a) subject 1 was neutral to changes in current dose, and the electric field intensity remained proportional to the injected current; (b) subject 2 had better focality from a dose of 2 mA or more (but not from 1 mA); and (c) subject 3 gained better stimulation from 1 mA compared to 2 or 3 mA of current dose. Such interindividual inconsistency in tDCS due to the current dose has been widely reported in previous studies [17,18,19,56,57,58,59,60,61,62]. With this i-SATA(MNI) framework post-processing the structural scans simulated in ROAST, tDCS users can configure personalized protocols for montage selection (refer to [11,35]) and identify the optimal current dose for cortical targeting (guided by DTDI). We applied the framework on a wide age range (20 to 80 years) of individuals from both sexes to highlight the importance of DTDI and the need for a focality-based selection of the current dose.

Previous tDCS based studies have combined electroencephalography, functional MRI, or transcranial magnetic stimulation to determine the current dose for optimal targeting [63,64,65,66]. Recently, a computational study has put forward the model to reverse calculate the current dose from the simulated electric field [16] based on the assumption that the intensity of current flow increases linearly with current dose [54]. A similar prototype was also put forward by Evans et al. [20]. All these studies used young, healthy subjects to delineate the model. On one hand, we see this linearity (constant value of DTDI) being followed in the MNI standard head model (Figure 1) and to an extent in young and middle-aged individuals (Figure 2A and Figure 3A). However, on the other hand, linearity appears to diminish with advancement in age (Figure 2B,C and Figure 3A), suggesting a potential nonlinear relationship.

The different values of DTDI as a function of current dose across different subjects could be because the injected current might become clustered in brain areas (referred to as hotspots), a phenomenon that has been widely reported in tDCS studies [6,25,26,67,68,69]. Hotspots cause shunting of the current towards the surrounding brain tissue and a surge in the electric field strength at localised areas [70]. Areas that form hotspots can be away from the electrode site as well [70]. In the two cases presented in Figure 2 (Subject 2 and 3), the superior parietal lobule appears to be a hotspot. Here, it is difficult to comprehend the neuroanatomical factors that contribute to the formation of such hotspots. It has been found that tissue heterogeneities and pathological alterations (like neurodegeneration and cerebral infarcts) are the primary contributors [26,70]. As we age, atrophy in the neural configuration escalates the nonlinearities in the spatial distribution of induced electric fields [71,72]. Care must be taken to limit the possibilities of such hotspots forming for clinical application of tDCS [73]. DTDI that considers the current density in target and non-target areas incorporates an understanding of the effect of hotspots to provide a rigorous estimate of the optimal current dose.

Since maximum stimulation might not be received at the target ROI, nor in a consistent location [6,17,18,74], the interindividual and intraindividual variation in DTDI can provide insights for the appropriate determination of current dose based on the age and sex of a healthy individual. In young adults, the focality of stimulation remains intact (approximately) across the doses, indicating that there is flexibility in choosing (individualizing) a dose depending on the extent of current density desired at the target ROI. However, the focality declines with advancing age (middle age onwards, see Figure 3). This decline is higher for males compared to females. Such sexual dimorphism in tDCS-related effects have been reported in previous studies [75] and several factors related to cortical anatomy like volume [76], bone density [77], hormonal levels [77], and electrode location have been postulated to account for it. We also found that higher current doses can enhance the focality in older adults. This is in support of a recent study [78] which reported that cerebral atrophy in older adults was causing the reduction in the amount of current reaching the target ROI. With that being said, we feel that it is important to guide our readers on how to use DTDI in their tDCS study (at an individual or group level), wherein their interest may lie in stimulating single or multiple target ROIs. We will discuss this in the next section.

### How to Use DTDI

(A). DTDI for Group-level Study

We have shown how DTDI can be used to titrate the current dose at the individual level. This can be performed at the group level also. Evans et al. [20] have suggested that the input current should be varied across individuals to maintain a constant current density at the target ROI. While we agree with them, we also suggest that the focality of the stimulation needs to be considered, especially when older individuals are recruited for the study. For primary clinical/therapeutic applications of tDCS, the focality, as revealed by DTDI, could be especially useful for setting tDCS dosage. Although the compatibility of the patient with the computationally recommended dose is always important [79], recent studies have indicated that participants readily tolerate tDCS current up to 4 mA [80,81].

In group studies in which researchers do not want to vary the current from subject to subject, DTDI values may still be used in two different ways to improve the efficacy of the study. The first would be to include a threshold for DTDI (e.g., DTDI ≥ 0.75) as a precautionary measure when individualizing the current dose. We recommend this value because studies pertaining to peripheral nerve stimulation have also found that such a threshold value (≥ 0.75) for the ‘selectivity index’ could ensure that the targeted region of the nerve is well activated [28,30]. While this may narrow down the suitability of subjects, such inclusion criteria could reduce the variability of tDCS. The second would be to use DTDI analyses for the populations under study to determine, at the start of the study, the optimal value of the tDCS current dose to be used on all subjects that will produce the greatest focality and the least amount of subject–subject variability in DTDI. For example, the current study suggests that for the F3-RSO montage, if you are including older and younger subjects, a higher current value (for the study overall) might produce the least variability in terms of focality of tDCS.

(B). DTDI for Multiple Target ROIs

We have shown how DTDI can be used for a target ROI. However, a user may be interested in stimulating multiple target ROIs, or a network of ROIs specific to a particular functionality [82,83,84,85,86,87,88]. This may be because studies have shown that more than one brain area can be involved in neuropsychiatric disorders like depression [89,90]. Alternatively (in an unexpected scenario), it may also happen that two or more ROIs receive the same current density after a montage is simulated for an individual’s brain. In such scenarios, the DTDI can be averaged across all the target ROIs ($\frac{DTDI\;for\;first\;TargetROI+DTDI\;for\;second\;TargetROI+..\ldots +DTDI\;for\;nth\;targetROI}{Number\;of\;areas\;\left(n\right)}\;$). The averaged DTDI will also lie in between 0 and 1. The optimal dose should focus on determining the current at which the value of the averaged DTDI exhibits an increasing trend.

## 5. Limitations

Here, we show how DTDI aids selection of focality-based current dose in tDCS. Although the findings on the relationship between tDCS dose, sex, and age could be limited by the small sample size, they reveal that, while brains which have a linear relationship between the electric field intensity and the tDCS current are flexible to the selection of current dose, brains that have a nonlinear relationship are dose sensitive. Therefore, biomarkers that could identify brains that can behave nonlinearly prior to the onset of an experiment could improve the precision of tDCS. However, this will require large datasets for validation of the biomarker. We leave it for future studies to investigate the exact nature of such nonlinearities and to determine the factors (structural, functional, and behavioural) that contribute to them.

Finally, we would like to highlight that DTDI can be estimated from i-SATA as well. The Talairach atlas space has several benefits [36], and similarly the i-SATA(MNI) that was developed for users who prefer the MNI space has its own advantages [43]. However, simulation in i-SATA(MNI) is considerably faster than in i-SATA. This is because both i-SATA(MNI) and the integrated SPM anatomy toolbox for cortical labelling are MATLAB-based and automated. This makes i-SATA(MNI) efficient at post-processing large data sets, a trend that is emerging in neuroscientific research.

## 6. Conclusions

The study extends the i-SATA framework to the MNI atlas space. With i-SATA(MNI), it will be easier to calculate the individualized dose as suggested in previous studies [15,16,20,21]. Here, we introduce the DTDI as a measure to titrate the individualized current doses and select the optimum dose that has high focality and could appropriately stimulate the target ROI in an individual. This will facilitate the personalized application of tDCS so that the desired stimulation benefits are achieved. Using a montage that has been found to be optimal for a DLPFC stimulation, DTDI analysis across a broad spectrum of men and women of different age groups has revealed that focality decreases with advancing age, especially in males with more than 40 years of age. Finally, the study reveals that the selection of a current dose that increases the focality is strictly necessary for older (> 60 years) individuals irrespective of sex.

## Figures and Tables

**Figure 1 jpm-11-00940-f001:**
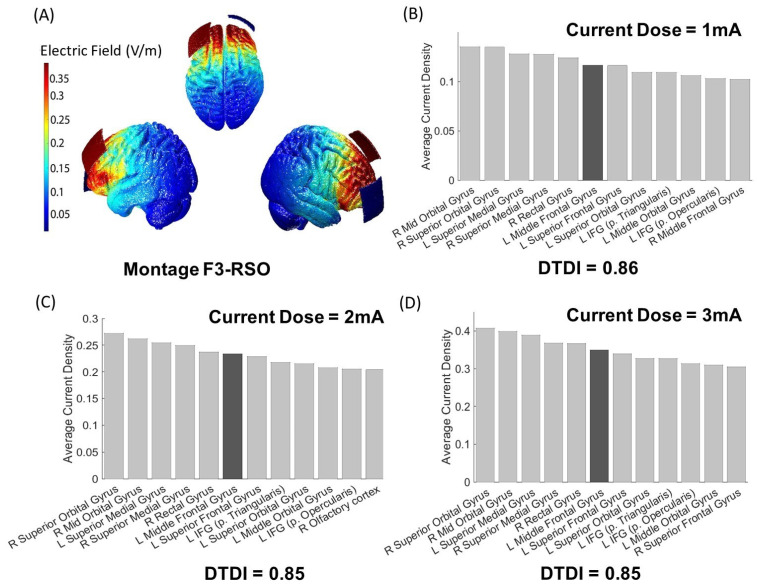
Illustration of the applied montage F3-RSO (shown in (**A**)) and output of i-SATA(MNI) for the MNI152 standard head image across the three current doses: (**B**) 1 mA; (**C**) 2 mA; and (**D**) 3 mA. The average current density at the target ROI (left middle frontal gyrus) is shown in the dark-gray colored bar. The DTDI index (~ 0.85) remains fairly constant across the three current doses, indicating a linear relationship between the injected current and induced electric field. For the standard head, tDCS users can choose any current dose, and their choice depends on the intensity of the desired stimulation in the target ROI. Note that the i-SATA(MNI) outputs shown in the figure highlight only the ROIs that received the highest current density (top 10%) amongst all other areas. A user can use the i-SATA(MNI) toolbox to obtain the average current density across all the brain regions.

**Figure 2 jpm-11-00940-f002:**
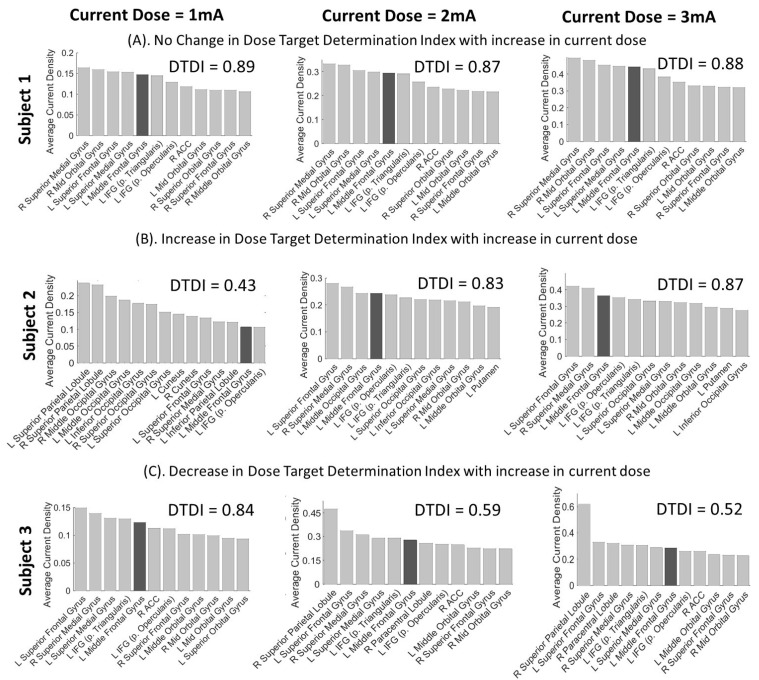
Illustration of the variation in DTDI for three current doses (1 mA, 2 mA, and 3 mA) with F3-RSO montage applied over three individuals showing: (**A**) no change in DTDI with an increase in current dose; (**B**) increase in DTDI with an increase in current dose; and (**C**) decrease in DTDI with an increase in current dose. Subject 1, with no change in DTDI, is neutral to variation in current dose, and the dose can be adjusted based on the intensity of the stimulation desired at the target ROI. Subject 2, showing increase in DTDI, would receive adequate stimulation from a higher dose (above 1 mA), whereas subject 3, showing a decrease, will most likely benefit from the lower dose (1 mA).

**Figure 3 jpm-11-00940-f003:**
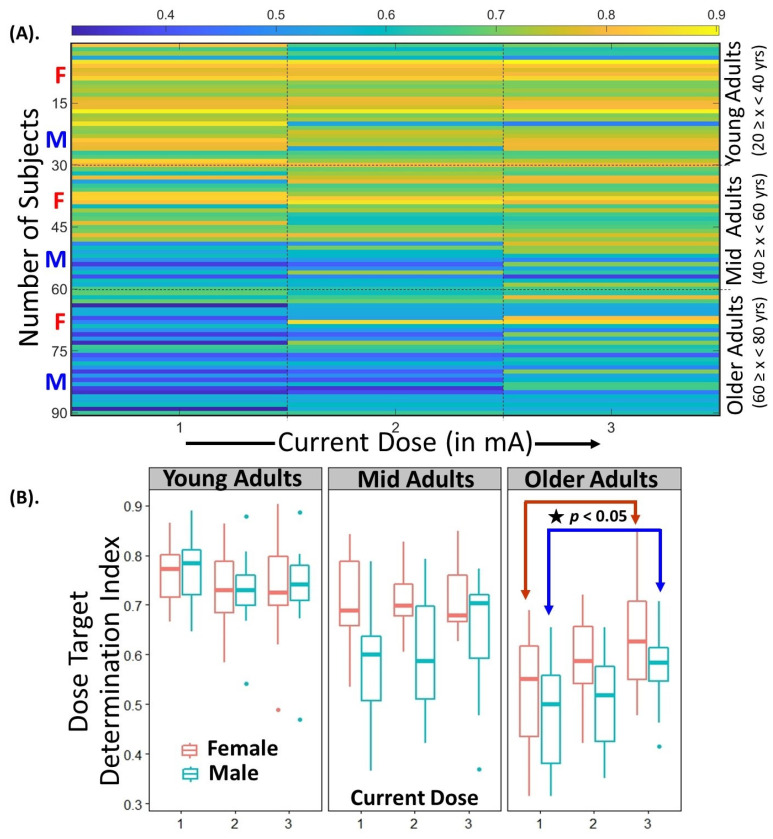
Illustration of the variation in DTDI at individual and group level with: (**A**) The individual variation of DTDI values (0 to 1) in the females (font in red ‘F’) and males (font in blue ‘M’) distributed equally across the three age groups ((i) young adults (20 ≤ x ˂ 40 years), (ii) mid adults (40 ≤ x ˂ 60 years), and (iii) older adults (60 ≤ x ˂ 80 years)) for the three current doses (1 mA, 2 mA, and 3 mA). The interindividual and intraindividual variation in DTDI clearly shows the current dose that could be appropriate for an individual to stimulate the target ROI after a montage has been fixed; (**B**) The variation of DTDI for both sexes across the three age groups using a three-way mixed ANOVA. The DTDI decreases with an increase in age. In mid and older adults, females show higher focality compared to males for the three current doses (1 mA, 2 mA, and 3 mA). In older adults only, the significant (*p* < 0.05, Bonferroni-corrected) difference between DTDI at 1 mA and 3 mA for both sexes conveys that higher current doses are required to appropriately stimulate the target ROI.

## Data Availability

We would like to convey our gratitude to the CAM-CAN team (https://camcan-archive.mrc-cbu.cam.ac.uk/dataaccess/) for providing access to the dataset.

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
