# Peer review of "Focality-Oriented Selection of Current Dose for Transcranial Direct Current Stimulation"

_jpm, 2021, doi:10.3390/jpm11090940_

Round 1
Reviewer 1 Report
In “Focality Oriented Selection of Current Dose for Transcranial 2 Direct Current Stimulation”, the authors present some advances in their i-SATA toolbox (the addition of the capability to register to MNI reference space), introduce a quantification metric for “focality of stimulation”, and then use their toolbox to make (physically implausible) predictions for patient-specific dose-recruitment relationships to electrical stimulation. While there is valuable underlying work which has been done by the authors, my interpretation is that the manuscript could be summarized as "the authors show their ROAST software makes a physically implausible prediction," which is my grounds for a recommendation of reject.
I am skeptical as to how well the DTDI metric introduced by the authors captures “focality” given that an applied stimulus resulting in nearly-uniform current density in the entire brain would evaluate to the same DTDI value as perfect focal stimulation of a single brain region. A brief review of other metrics proposed in the field needs to be included for comparison with the author’s DTDI metric.
As DTDI is in effect a ratio metric, if electric field intensity is proportional to current (at the ROI or in any other parcel of brain) then DTDI should be insensitive to dose. It’s not outright impossible for electric field nonlinearities (e.g. field-strength-dependent conductivities) to cause deviations from this but given that this is a simulation study the exact nature and importance of those nonlinearities need to be documented and it needs to be shown that when those nonlinearities are excluded from simulation the expected linear result arises. In effect the claim of nonlinearities is an extraordinary claim requiring extraordinary evidence.
The authors claim determining patient-specific dose is more important for men > 40 years of age (based on a relatively small sample). Is this due to a greater distance from the reference space to the patient space for this cohort? This comparison should be quantified for each of the groups in the study.
What, if any, are the advantages of using the MNI reference space over the Talairach atlas space?
The citations are slipshod – claims are cited which are not backed up by the works cited (e.g. [15-16])
Reviewer 2 Report
The manuscript is potentially interesting. However, I have a few comments:
- Please indicate the full statistical values and test, including exact p-value, f score, t score, and effect size.
- Please indicate the number of participants in each group included in the study and their health status. Currently, it is not mentioned at all.
- The claims about "The focality decreases with age and the decline is stronger in males. Higher current dose at older age can enhance the focality of stimulation" cannot be simply supported by t-test. Please indicate any correlations.
- In the discussion, please discuss potential limitations as well as comparison between other methods including TMS and DBS.
Reviewer 3 Report
Thank you for the opportunity to review this well-written work. The authors have explicitly described Dose-Target-Determination-Index for quantification of tDCS focality and evaluated the relationship between current dose and focality. This is novel and important area of research. I have couple of minor comments:
- Page 2 Ln 89,91, 92: Please replace the term "intensity" with "density" in these sentences and in the formula for individual dose as 0.25 and 0.5 are in mA/m2.
- Page 9 Ln 296: The authors have referred to two studies in this statement. Please rephrase the sentence.
Author Response
Dear Reviewer,
We thank you for the close read of this manuscript and insightful suggestions.
We are submitting our revised manuscript (No: jpm-1289513) entitled “Focality Oriented Selection of Current Dose for Transcranial Direct Current Stimulation” to address all the raised concerns.
Please find below the detailed responses (in blue) to the reviewer comments (in italics). For convenience, changes to the manuscript are quoted verbatim (normal font) when appropriate. We believe you will find the corrected manuscript much improved and suitable for publication.
Reviewer #3
Thank you for the opportunity to review this well-written work. The authors have explicitly described Dose-Target-Determination-Index for quantification of tDCS focality and evaluated the relationship between current dose and focality. This is novel and important area of research. I have couple of minor comments:
(R3Q1). Page 2 Ln 89,91, 92: Please replace the term "intensity" with "density" in these sentences and in the formula for individual dose as 0.25 and 0.5 are in mA/m2.
We thank the reviewer for this suggestion. We have replaced ‘intensity’ with ‘density’ in the following lines. For the convenience of the reviewer, updates to the manuscript have been pasted below.
Page 2 (Introduction, Line 89-92)
Suppose the calculated current density at the target ROI is 0.25 mA/m2 when 1 mA of current is applied on the scalp. To achieve a desired density of 0.5 mA/m2 at the target ROI, the required dosage (individualized) can be reverse calculated as = 2 mA].
(R3Q2). Page 9 Ln 296: The authors have referred to two studies in this statement. Please rephrase the sentence.
We thank the reviewer for pointing this out. We have now referred to the study that is most relevant study and have removed the other study. The change in the manuscript can be read as follows.
Page 9 (Discussion, Line 296)
Recently, a computational study had put forward the model to reverse calculate the current dose from the simulated electric field [16] based on the assumption that the intensity of current flow increases linearly with current dose [50].

Round 2
Reviewer 1 Report
The authors did not adequately respond to the requested revisions highlighting significant flaws in the proposed work, instead claiming that all such revisions to address the significant concerns presented are "beyond the scope of current work".
Regarding the originality of the proposed metric, a large number of selectivity metrics are known in the nerve stimulation literature.
Author Response
Dear Reviewer,
We are submitting our revised manuscript (No: jpm-1289513) entitled “Focality Oriented Selection of Current Dose for Transcranial Direct Current Stimulation” to address the concerns raised.
We thank you for the close read of this manuscript and insightful input. In particular, highlighting that a metric similar to DTDI exist in the field of nerve stimulation has helped to improve the work.
Please find below the detailed responses (in blue) to the comments (in italics). For convenience, changes to the manuscript are quoted verbatim (normal font) when appropriate. We believe you will find the corrected manuscript much improved and suitable for publication.
(R1Q1). The authors did not adequately respond to the requested revisions highlighting significant flaws in the proposed work, instead claiming that all such revisions to address the significant concerns presented are "beyond the scope of current work".
We agree with the reviewer that one of the revisions suggested previously could not be addressed. For a quick glimpse, the question is pasted below.
(PREV-R1Q3). As DTDI is in effect a ratio metric, if electric field intensity is proportional to current (at the ROI or in any other parcel of brain) then DTDI should be insensitive to dose. It is not outright impossible for electric field nonlinearities (e.g. field-strength-dependent conductivities) to cause deviations from this but given that this is a simulation study the exact nature and importance of those nonlinearities need to be documented and it needs to be shown that when those nonlinearities are excluded from simulation the expected linear result arises. In effect the claim of nonlinearities is an extraordinary claim requiring extraordinary evidence.
The difficulty to address this is because we exactly cannot determine the multiple factors and their interplay that contributes to the observed non-linearity. Brain being a complex and convoluted structure, we need to extract the various parameters that include -volume (Cerebrospinal fluid, White matter, and Grey matter), thickness, surface area, Torque and many more at the whole-brain (global) level. This won’t be sufficient, and we may need to narrow down all these parameters to the regional-level as well. Evaluating so many parameters (along with the existing content) may confuse/deviate the readers. Moreover, unlike peripheral nerve stimulation wherein such detail investigation are approachable, in the field of brain stimulation, it is difficult and awaits further/fresh investigations.
(R1Q4). Regarding the originality of the proposed metric, a large number of selectivity metrics are known in the nerve stimulation literature.
We thank the reviewer to providing the direction (nerve stimulation literature) about existence of such metrics and its widespread usage. This has enhanced our work. We have included the literatures about selectivity index (as mentioned in nerve stimulation) throughout the manuscript (introduction, methodology and discussion). Changes in the manuscript read as follows.
Page 2 (Introduction, Line 75-78)
At this point it is important to mention that other stimulation techniques (like peripheral nerve stimulation) are also motivated to increase the stimulation intensity at tar-get region while minimizing the stimulation received at non-target regions [27–30]. With tDCS, poor focality in stimulating the target region has constrained its efficacy.
Page 3 (Introduction, Line 110-112)
A similar metric defined as ‘selectivity index’ that measures the recruitment of the targeted region compared to other non-targeted regions is used to quantify the effectiveness in peripheral nerve stimulation [27–30].
Page 5 (Method, Section 2.4 Line 200)
A DTDI value = 0 will indicate no stimulation of target ROI.
Page 10 (Discussion, Line 359-362)
We suggest this value because studies pertaining to peripheral nerve stimulation have also found that such a threshold value (≥ 0.75) for ‘selectivity index’ could ensure that targeted region of the nerve is well activated [28,30].

Reviewer 2 Report
The authors have done a great job responding to my comments. I have no further comments.
Author Response
We thank the reviewer for the encouragement to our work